# Sample-efficient Antibody Design through Protein Language Model for Risk-aware Batch Bayesian Optimization

**Yanzheng Wang**
University of Bristol

**Boyue Wang**
University of Wisconsin - Madison

**Tianyu Shi** *
University of Toronto

**Jie Fu**
Hong Kong University of Science and Technology

**Yi Zhou**
BioMap (Beijing) Intelligence Limited

**Zhizhuo Zhang**[†]
BioMap (Beijing) Intelligence Limited

## Abstract

Antibody design is a time-consuming and expensive process that often requires extensive experimentation to identify the best candidates. To address this challenge, we propose an efficient and risk-aware antibody design framework that leverages protein language models (PLMs) and batch Bayesian optimization (BO). Our framework utilizes the generative power of protein language models to predict candidate sequences with higher naturalness and a Bayesian optimization algorithm to iteratively explore the sequence space and identify the most promising candidates. To further improve the efficiency of the search process, we introduce a risk-aware approach that balances exploration and exploitation by incorporating uncertainty estimates into the acquisition function of the Bayesian optimization algorithm. We demonstrate the effectiveness of our approach through experiments on several benchmark datasets, showing that our framework outperforms state-of-the-art methods in terms of both efficiency and quality of the designed sequences. Our framework has the potential to accelerate the discovery of new antibodies and reduce the cost and time required for antibody design.

## 1 Introduction

Antibodies, also known as immunoglobulins, are proteins produced by the immune system to recognize and neutralize foreign substances. They play a critical role in the body's defence against infections and diseases [27]. The variable regions of an antibody are responsible for antigen recognition, are highly diverse, and consist of three complementarity-determining regions (CDRs) named CDR1, CDR2, and CDR3. Among these CDRs, CDR3 exhibits the greatest variability and is often referred to as the "hypervariable" region [32]. Efficient antibody design is becoming more and more important because it has the potential to accelerate the development of effective treatments and vaccines [15, 13].

Throughout the antibody design process, we strive to harness the full potential of antibodies by tailoring their properties to meet specific requirements. By optimizing their affinity, stability, and

---

*Primary Corresponding Author, email: ty.shi@mail.utoronto.ca
[†]Secondary Corresponding Author, email:zzz2010@gmail.com

NeurIPS 2023 AI for Science Workshop.

other attributes, these designed antibodies offer promising prospects for targeted therapy, diagnostics, and various biomedical applications [18, 2].

Typically, Experimental antibody design and screening can be time-consuming and expensive. Simulation allows researchers to test a large number of potential antibody structure candidates and select the most promising candidates for further experimental validation, saving time and resources. Improving the process of simulations [33] can further provide insight into the properties and behaviour of antibodies, such as binding affinity and specificity, which may be difficult to determine experimentally [16, 8]. However, the sheer number of possible CDRH3 sequences in a combinatorial space makes it infeasible to exhaustively examine any antibody simulation framework [19]. Therefore, we need computational tools to guide our exploration of the protein landscape

Recently, Bayesian optimization has demonstrated its efficiency in exploring the sequence design space [16, 3]. Bellamy et al [6] compared how noise affects different batched Bayesian optimization techniques and introduced a retest policy to mitigate the effect of noise. Wang et al [30] discussed using Bayesian optimization (BO) to design chemical-based products and functional materials, showing that BO can significantly reduce the number of experiments required compared to traditional approaches. However, for antibody sequence design where the search space dimension is extremely large, it is very ineffective for Bayesian optimization. The choice of the acquisition function used to guide the optimization process can also impact its effectiveness, and there may be a trade-off between exploration and exploitation that must be carefully balanced.

We propose GLMAb-BO, an efficient way for antibody sequence optimization to address the above challenges. Our main contributions are improving exploration efficiency by using protein language models to filter out mutants with low fitness scores and designing a risk-aware acquisition function based on the uncertainty of the prediction to improve the explorer's ability. We demonstrate the effectiveness of our proposed method on multiple antibody datasets. Our model can identify the sequence with the best fitness score in the fewest rounds compared to other baselines.

## 2   Related work

Specially, we can use fitness scores to evaluate the bio function of the sequence, which play a crucial role in antibody design as they serve as important indicators of the functional and structural quality of antibodies. Higher fitness scores generally indicate better binding affinity, stability, and other desirable properties. Many novel frameworks have been proposed to model various protein sequences. Especially for pre-trained language models which demonstrate transfer learning ability to predict fitness scores [29, 21]. In the context of antibody design, predicting fitness scores can be highly beneficial. It provides a cost-effective alternative to conducting time-consuming and expensive wet-lab experiments. By utilizing computational models and machine learning techniques, researchers can efficiently evaluate the fitness of a large number of antibody sequences, prioritizing those with higher predicted fitness scores for further experimental validation. The need for better exploration algorithms, such as batch Bayesian optimization (BO), has gained attention in addressing the challenges of sequence design. Belanger et al [5] explored the application of batched Bayesian optimization in the context of biological sequence design, addressing the unique challenges and investigating design choices for robust and scalable design. Furthermore, Gonzalez et al [10] proposed a heuristic method based on an estimate of the function's Lipschitz constant to capture the interaction between evaluations in a batch. A penalized acquisition function is used to collect batches of points, minimizing non-parallelizable computational effort. Khan et al [16] used a CDRH3 trust region to restrict the search to sequences with favourable developability scores.

These studies highlight the ongoing efforts to address the challenges in sequence design for antibody engineering. By incorporating bayesian optimization, researchers aim to enhance the efficiency and effectiveness of antibody design and improve the sequence diversity.

## 3   Problem Formulation and Background

### 3.1   Antibody Sequence Design

Antibody Sequence Design can be formulated as a constrained optimization problem [1, 31, 16, 22]. Let $x$ be a vector representing the CDRH3 amino acid sequence, and let $f(x)$ be a fitness function that

quantifies the quality of the antibody sequence in terms of target specificity and developability. The problem is to find the optimal sequence $x^*$ that maximizes the scoring function subject to constraints:

$$\max_x f(x) \text{ s.t. } x \in \mathcal{X}, \, g(x) \leq 0, \tag{1}$$

where $\mathcal{X}$ is the set of all possible amino acid sequences for the CDRH3 region and $g(x)$ represents constraints on the biophysical properties of the sequence, such as stability and solubility. The optimization problem aims to find the best antibody sequence that satisfies the biophysical constraints and has the highest target specificity and developability scores. Bayesian optimization methods can be used to efficiently solve this optimization problem by iteratively proposing candidate sequences that are subsequently evaluated by a surrogate model and passed to an acquisition function that balances exploration and exploitation.

### 3.2 Bayesian optimization

Bayesian Optimization (BO) is a sequential model-based optimization technique used to solve expensive black-box optimization problems with a limited budget of function evaluations, which has been applied to sequence modelling [16, 30].

We can express the BO process as follows: Let $f(x)$ be the unknown fitness function we aim to optimize, where $x \in \mathcal{X}$ is the input variable. Our goal is to find the global optimum $x^*$ that maximizes $f(x)$. However, doing a wet lab experiment to evaluate $f(x)$ is expensive and time-consuming. The acquisition function, denoted by $\alpha(x)$, measures the utility of evaluating a point $x$ based on the current surrogate model. $\alpha(x)$ balances exploration and exploitation by favouring points with high uncertainty (exploration) or high expected improvement (exploitation). Popular acquisition functions include expected improvement (EI), upper confidence bound (UCB), and probability of improvement (PI) [34, 16].

The next evaluation point is selected by optimizing the acquisition function over the input space $\mathcal{X}$:

$$x_{n+1} = \text{argmax}_{x \in X} \alpha(x) \tag{2}$$

After evaluating $f(x_{n+1})$, we update the surrogate model with the new observation $(x_{n+1}, y_{n+1})$ and repeat the process until the budget of function evaluations is exhausted or a satisfactory solution is found. Batch BO improves this by minimizing the exploration rounds.

## 4 Method

### 4.1 General language model guided candidate pool generation

Intuitively, we propose to use the General language model (GLM) trained on diverse antibody datasets to score the candidate pool and filter out the sequence with lower fitness values in the vast sequence space. Let $\mathcal{C}$ be the candidate pool consisting of $N$ protein sequences, and let $f(x_i)$ be the fitness score of sequence $x_i$ from candidate pool $\mathcal{C}$ obtained from the protein language model. We determine the threshold fitness score $t$ that filters out $m\%$ of the sequences with fitness scores less than or equal to $t$. In the process of training our protein model GLM-Ab, we randomly mask one or two of the CDR regions by replacing the entire region with a random mask. We also conduct random mask fragments, by randomly masking one or more sections of the sequence.

Then, we can use GLM-Ab to score the sequences and determine an index $k$ such that $f(x_k) \leq t < f(x_{k-1})$. Furthermore, by setting $t = f(x_k)$, the filtered set of sequences $\mathcal{C}'$ with small search space and higher naturalness is obtained as:

$$\mathcal{C}' = x_i \in \mathcal{C} \mid f(x_i) \geq t \tag{3}$$

In other words, $\mathcal{C}'$ contains all sequences in $\mathcal{C}$ with fitness scores greater than or equal to $t$ based on GLM scoring.

## 4.2 Risk aware Bayesian optimization

Many previous works have been proposed to leverage uncertainty for biological discovery and sequence design [12, 34, 16]. However, using traditional Gaussian processes [12] to measure the uncertainty for a large sequence is extremely inefficient. Due to the complexity of the antibody design space, we propose a risk-aware exploration to balance exploration and exploitation by selecting points with high expected improvement and lower risk. In each round of optimization, we train an ensemble of models to estimate the uncertainty, similar to the approach taken by PEX [22].

We assume the output of $M$ surrogate models follows a normal distribution $\mathcal{N}(\mu_s, \sigma_s)$. We can divide the uncertainty of those model predictions as epistemic uncertainty (EU), $\sigma_e^2$; and aleatoric uncertainty (AU), $\sigma_a^2$ [25, 28]:

$$\sigma_e^2 = \frac{1}{M} \sum (\mu - \mu_s)^2, \quad \sigma_a^2 = \frac{1}{M} \sum_s \sigma_s^2(x), \tag{4}$$

where EU is based on the variance between the predictions of different surrogate models, and the AU-estimated standard deviation provides a measure of the uncertainty associated with the predicted values. EU quantifies the uncertainty associated with the lack of knowledge or variability in the models themselves. EU can be reduced by increasing the number or quality of models.

Unlike PEX, we use a UCB acquisition function to evaluate sequence $x$. The UCB acquisition function is defined as:

$$\alpha(x) = \mu(x) + \beta\sigma(x), \tag{5}$$

where $\mu(x)$ is the mean ensemble prediction generates from surrogate models for a sequence $x$, and $\beta$ is a hyperparameter ( as 0.5 in our experiment) that controls the trade-off between exploration and exploitation, and $\sigma(x)$ is the ensemble standard deviation function of the surrogate model for sequence $x$. In other words, $\sigma(x)$ represents the aleatoric uncertainty of the prediction for sequence $x$.

The risk-aware modification based on Equation 5 introduces a penalty term that depends on the aleatoric uncertainty of the fitness values in the candidate pool:

$$\alpha_{risk}(x) = \mu(x) + \frac{\beta}{m + risk}\sigma(x) \tag{6}$$

where $risk$ is the parameter that measures the variability, i.e., epistemic uncertainty, of the fitness values prediction based on the surrogate model for the whole candidate pool. we select $m = 0.5$ is a constant to avoid dividing by a very small value. The general purpose of this acquisition function is to discourage the selection of points with high variability, which can lead to unstable and unpredictable performance. To be more specific, the measure for risk is defined as:

$$risk = \frac{1}{|C|} \sum_{i=1}^{|C|} \sigma_i \tag{7}$$

It is calculated as the average of aleatoric uncertainty for the fitness values evaluation in the whole generated candidate pool, where $C'$ is the filtered candidate pool, and $\sigma_i$ is the standard deviation of the fitness values prediction for the $i^{th}$ candidate sequence.

In each round, we train the surrogate model $f_\theta$ on the queried sequences with true fitness scores from wet lab experiments (same as [22]). In the first few rounds, the surrogate model lacks good prediction ability for the candidate pool and could have a higher epistemic uncertainty [25]. The rationale for the risk measure is to consider epistemic uncertainty for the whole candidate pool, which indicates a high risk of selecting a suboptimal point that may lead to a performance drop.

## 4.3 GLMAb-BO

The full algorithm of our proposed algorithm can be found in Algorithm 1. In each round of black-box optimization, the whole framework is required to generate a query batch based on the measured

---
**Algorithm 1** Risk-aware Bayesian Sequence optimization
---
**Input**: Starting sequence $(s^{wt}, f(s^{wt}))$, Pre-trained protein language model $\mathcal{G}$, surrogate model $f_\theta$, measured buffer $\mathcal{D}$, whole candidate pool $\mathcal{C}$

**Parameter**: Initialize model parameter $\theta$
---
 1: **for** $t = 1$ to $T$ **do**
 2:     **while** condition **do**
 3:         Use Equation 3 to generate filtered sequence pool $\mathcal{C}'$ with higher naturalness.
 4:         Train ensemble of surrogate models $f_\theta$ to get prediction $\mu$ and uncertainty $\sigma$, and $risk$.
 5:         Use the acquisition function based on Equation 6 scoring $\mathcal{C}'$ to generate query sequence batch $D_t^{query}$.
 6:         Measure ground-truth fitness of $D_t^{query}$ by wet-lab experiments.
 7:         Update Surrogate model $f_\theta$ using $D_t^{query}$.
 8:     **end while**
 9: **end for**
---

fitness score through wet lab experiments. We first utilize the pre trained unsupervised GLM-Ab model to narrow down the candidate pool sequence space. Then, we integrate risk-aware batch Bayesian optimization to propose a query batch for web lab experiments. The visualization of the whole framework is in Figure 1.

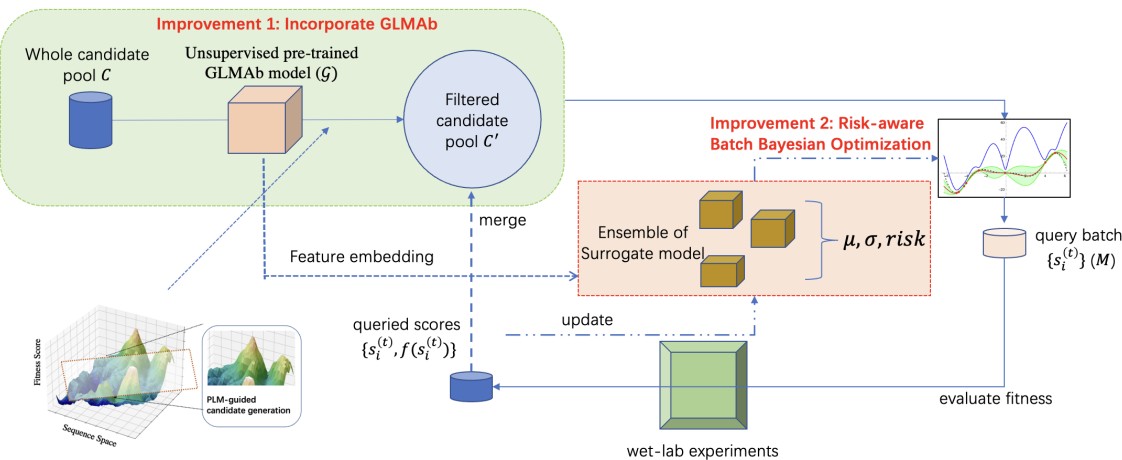

Figure 1: Framework overview. In our proposed GLMAb-BO framework, we first use the pre-trained GLM-Ab model $\mathcal{G}$ to filter out the sequence with unsatisfying naturalness in the candidate pool and acquire $\mathcal{D}'$, then we train an ensemble of surrogate models with GLM-Ab's feature encoding to predict the fitness the remaining sequences. When we acquire the ensemble mean $\mu$, prediction standard deviation $\sigma$, and the $risk$, we utilize the proposed risk-aware Bayesian Optimization (BO) acquisition function to further evaluate the sequences. Finally, we use the top 100 sequences with high predicted naturalness to conduct a wet-lab experiment (we use a hypothetical scenario due to time constraints for replacement in this study) and perform another round of exploration until we reach the exploration rounds limits.

## 5 Experiments

Absolut! framework [23] is used as a computational alternative to wet lab experiments for generating antibody-antigen binding datasets. It provides a deterministic simulation of binding affinity using coarse-grained lattice representations of proteins, allowing evaluation of all possible binding conformations between a CDRH3 sequence and an antigen. The framework has been benchmarked and shown to produce consistent results compared to experimental data [16, 14]. And we use this framework to generate the initial whole candidate pool.

## 5.1 Baseline methods

In this study, several methods for antibody design optimization are compared. The **Combinatorial Bayesian Optimization for Antibody Design (antbo)** [16] approach employs combinatorial Bayesian optimization to efficiently design antibody CDRH3 regions, using a trust region and a black-box oracle for scoring specificity and affinity. **Proximal Exploration (pex)** [22] introduces the Proximal Exploration algorithm and the Mutation Factorization Network architecture, which prioritize high-fitness mutants with low mutation counts for protein sequence design. The **Batch Bayes Optimization (batchbo)** [5] method uses a neural network ensemble with uncertainty estimates to guide sequence batch selection using expected improvement. **Random Search** is employed as a baseline for method comparison, randomly selecting subsets of sequences for reference. These diverse methods provide insights into the optimization landscape and guide the development of more advanced algorithms for protein sequence design.

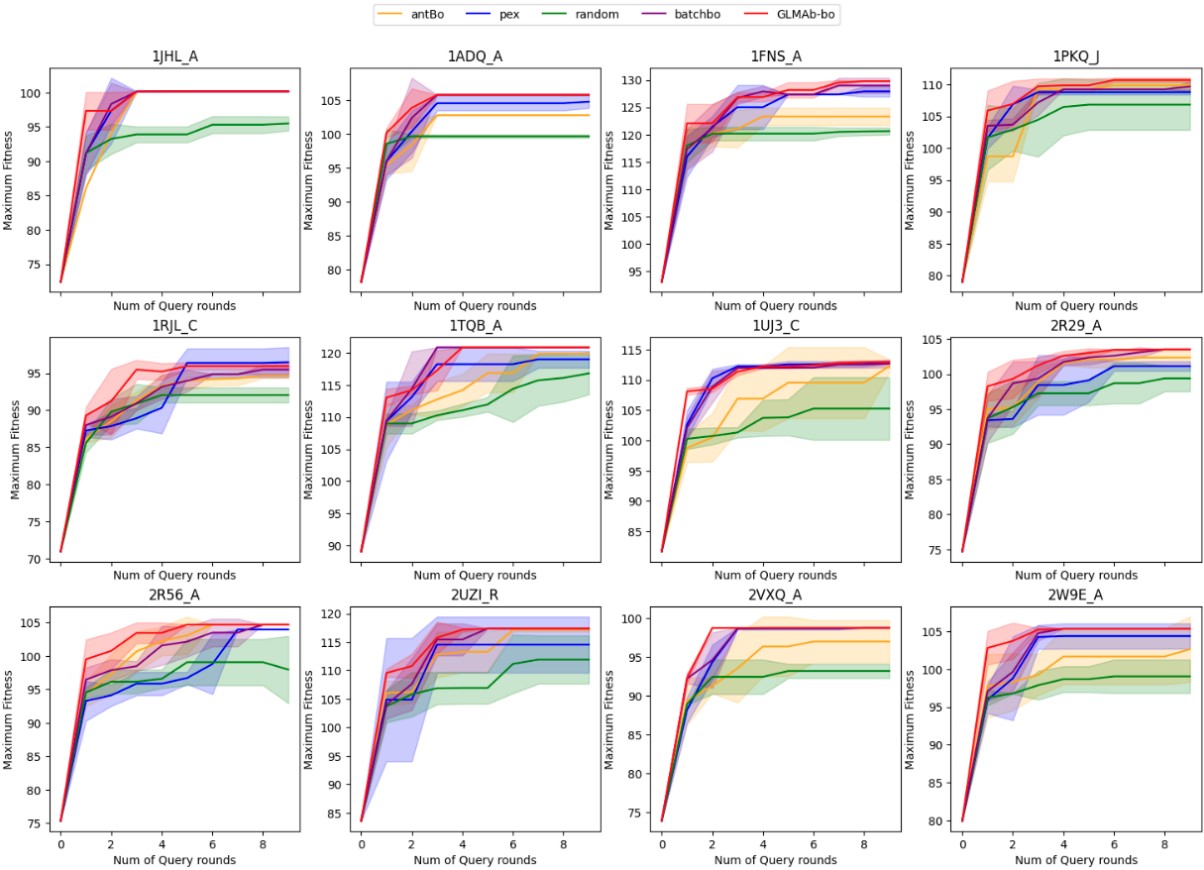

Figure 2: Experimental results comparison on antibody datasets, each round of black-box optimization can generate 100 proposal sequences. We use maximum measured fitness in each round as the evaluation metric. The shaded area indicates the standard deviation given 5 random seeds.

## 5.2 Ablative study methods

In the ablative study, we assess the effectiveness of our proposed enhancements in the GLMAb-BO method through various ablations. These include **GLMAb-score**, which focuses solely on the highest predicted score from GLMAb on the candidate pool, and **GLMAb-select**, which removes the acquisition function and relies solely on the surrogate model for sequence selection. Additionally, **GLMAb-random** eliminates both the acquisition function and surrogate model, utilizing the GLM model to filter sequences and then randomly selecting the top 100. **GLMAb(w/o emb)-BO** removes the embedding of GLMAB's CNN surrogate model to evaluate the feature embedding module.

Table 1: Comparison of sequence optimization results on different datasets, we summarized maximum fitness over 5 rounds, 10 rounds, and average maximum fitness over 10 rounds

| Method | 1JHL_A | 1ADQ_A | 1FNS_A | 1PKQ_J | 1RJL_C | 1TQB_A | 1UJ3_C | 2R29_A | 2R56_A | 2UZI_R | 2VXQ_A | 2W9E_A | overall |
|---|---|---|---|---|---|---|---|---|---|---|---|---|---|
| antbo (10) | 100.18 | 102.76 | 123.32 | 109.86 | 94.64 | 119.84 | 112.26 | 102.34 | 104.69 | 117.17 | 97.01 | 102.62 | 107.22 |
| pex (10) | 100.18 | 104.75 | 127.91 | 108.79 | 96.47 | 118.98 | 112.89 | 101.11 | 103.97 | 114.54 | 98.77 | 104.37 | 107.73 |
| random (10) | 95.49 | 99.65 | 120.64 | 106.83 | 92.05 | 116.79 | 105.26 | 99.37 | 99.06 | 111.91 | 93.21 | 99.04 | 103.28 |
| batchbo (10) | 100.18 | 105.76 | 128.97 | 109.69 | 95.48 | 120.84 | 112.67 | 103.50 | 104.69 | 117.38 | 98.77 | 105.34 | 108.61 |
| GLMAb-BO (10) | 100.18 | 105.76 | 129.78 | 110.70 | 95.95 | 120.84 | 112.89 | 103.50 | 104.69 | 117.38 | 98.77 | 105.34 | **108.82** |
| antbo (5) | 94.07 | 98.38 | 120.23 | 98.71 | 88.49 | 111.09 | 100.50 | 95.19 | 97.29 | 106.22 | 91.32 | 98.32 | 99.98 |
| pex (5) | 97.36 | 100.36 | 121.45 | 106.87 | 87.86 | 113.09 | 110.23 | 93.60 | 94.11 | 104.87 | 94.34 | 98.77 | 101.91 |
| random (5) | 93.25 | 99.65 | 120.17 | 102.91 | 89.80 | 109.00 | 100.73 | 95.27 | 96.14 | 105.82 | 92.46 | 96.78 | 100.16 |
| batchbo (5) | 98.34 | 102.36 | 121.20 | 103.68 | 89.12 | 114.39 | 108.84 | 98.66 | 97.86 | 107.60 | 94.63 | 99.63 | 103.03 |
| GLMAb-BO (5) | 97.34 | 103.84 | 122.07 | 106.93 | 91.21 | 114.19 | 108.52 | 99.30 | 100.76 | 110.77 | 98.77 | 103.73 | **104.79** |
| antbo (avg) | 95.37 | 99.13 | 119.18 | 104.35 | 90.21 | 112.91 | 104.51 | 97.64 | 99.17 | 105.39 | 92.89 | 98.62 | 102.03 |
| pex (avg) | 96.21 | 100.64 | 121.84 | 104.89 | 90.73 | 114.14 | 108.22 | 96.21 | 96.18 | 109.51 | 94.74 | 100.51 | 102.82 |
| random (avg) | 92.00 | 97.40 | 117.36 | 102.86 | 88.95 | 110.34 | 101.26 | 95.15 | 95.29 | 106.08 | 90.63 | 96.43 | 99.48 |
| batchbo (avg) | 96.30 | 101.69 | 122.79 | 104.92 | 90.68 | 115.88 | 107.83 | 98.31 | 98.82 | 111.30 | 95.21 | 101.34 | 103.76 |
| GLMAb-BO (avg) | 96.83 | 102.25 | 123.63 | 106.40 | 92.19 | 115.86 | 108.50 | 99.30 | 100.60 | 112.38 | 95.63 | 102.37 | **104.66** |

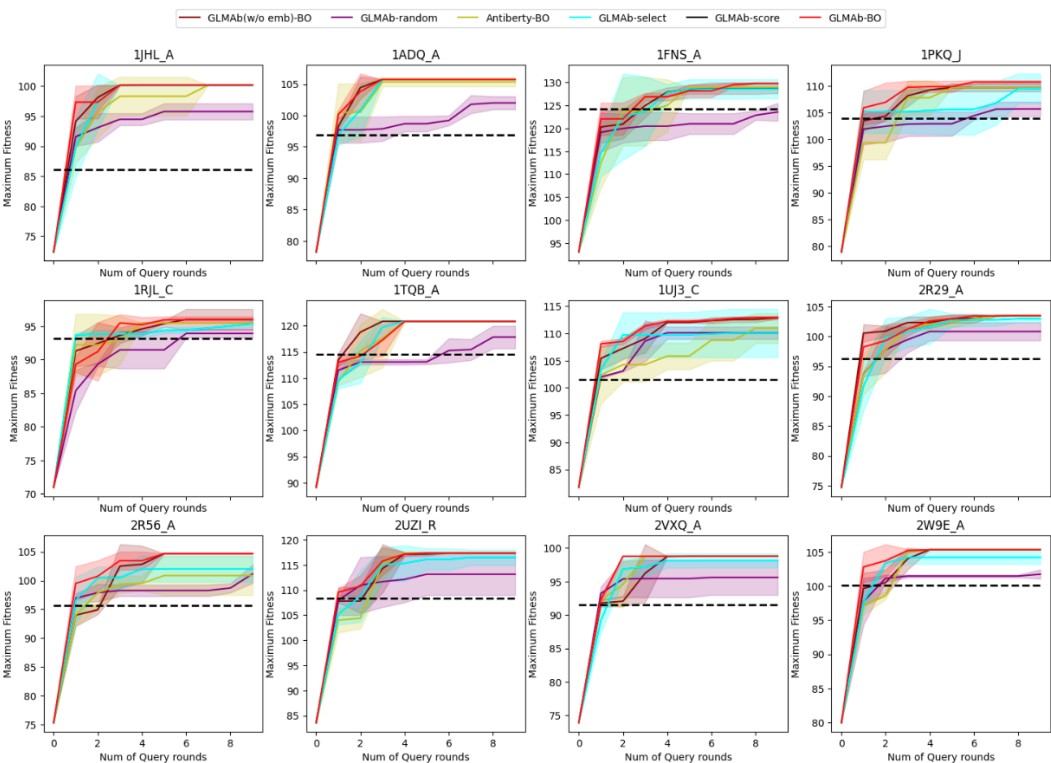

Figure 3: Ablative study experimental results comparison on antibody datasets with 5 random seeds.

Table 2: Ablation results on different datasets,we summarized maximum fitness over 5 rounds, 10 rounds, and average maximum fitness over 10 rounds.

| Method | 1JHL_A | 1ADQ_A | 1FNS_A | 1PKQ_J | 1RJL_C | 1TQB_A | 1UJ3_C | 2R29_A | 2R56_A | 2UZI_R | 2VXQ_A | 2W9E_A | overall |
|---|---|---|---|---|---|---|---|---|---|---|---|---|---|
| GLMAb(w/o emb)-BO (10) | 100.18 | 105.76 | 128.78 | 109.66 | 96.01 | 120.84 | 112.89 | 103.50 | 104.69 | 117.38 | 98.77 | 105.34 | 108.65 |
| GLMAb-random (10) | 95.78 | 102.01 | 123.56 | 105.67 | 93.92 | 117.82 | 110.10 | 100.85 | 101.15 | 113.15 | 95.61 | 101.74 | 105.11 |
| Antiberty-BO (10) | 100.18 | 105.34 | 128.97 | 109.64 | 95.48 | 120.83 | 110.94 | 103.50 | 100.88 | 117.38 | 98.77 | 105.34 | 108.10 |
| GLMAb-select (10) | 100.12 | 105.76 | 128.57 | 109.47 | 95.34 | 120.84 | 110.05 | 102.99 | 102.06 | 116.48 | 98.11 | 104.21 | 107.83 |
| GLMAb-BO (10) | 100.18 | 105.76 | 129.78 | 110.70 | 95.95 | 120.84 | 112.89 | 103.50 | 104.69 | 117.38 | 98.77 | 105.34 | **108.82** |
| GLMAb(w/o emb)-BO (5) | 98.11 | 104.45 | 120.94 | 104.34 | 92.43 | 118.78 | 107.23 | 100.91 | 94.93 | 107.81 | 92.06 | 100.50 | 103.54 |
| GLMAb-random (5) | 93.09 | 97.70 | 119.98 | 102.44 | 89.32 | 113.06 | 103.08 | 97.80 | 97.91 | 110.96 | 95.38 | 101.13 | 101.82 |
| Antiberty-BO (5) | 95.60 | 100.51 | 123.69 | 99.35 | 92.23 | 115.20 | 104.25 | 97.80 | 97.89 | 104.40 | 94.56 | 98.58 | 102.00 |
| GLMAb-select (5) | 97.45 | 100.89 | 122.53 | 105.18 | 93.84 | 112.97 | 109.71 | 99.84 | 100.49 | 108.74 | 96.87 | 103.37 | 104.32 |
| GLMAb-BO (5) | 97.34 | 103.84 | 122.07 | 106.93 | 91.21 | 114.19 | 108.52 | 99.30 | 100.76 | 110.77 | 98.77 | 103.73 | **104.79** |
| GLMAb(w/o emb)-BO (avg) | 96.59 | 102.12 | 123.06 | 105.24 | 92.23 | 116.72 | 107.78 | 99.73 | 99.30 | 111.74 | 94.66 | 101.62 | 104.23 |
| GLMAb-random (avg) | 92.49 | 97.39 | 118.25 | 101.34 | 89.57 | 111.92 | 105.60 | 97.08 | 96.15 | 109.21 | 93.13 | 98.94 | 100.92 |
| Antiberty-BO (avg) | 95.15 | 101.67 | 122.28 | 104.15 | 92.12 | 115.61 | 104.36 | 98.55 | 97.05 | 111.15 | 95.21 | 101.27 | 103.21 |
| GLMAb-select (avg) | 96.05 | 101.63 | 122.45 | 103.68 | 92.01 | 115.65 | 106.40 | 98.29 | 98.53 | 110.97 | 94.58 | 101.05 | 103.44 |
| GLMAb-BO (avg) | 96.83 | 102.25 | 123.63 | 106.40 | 92.19 | 115.86 | 108.50 | 99.30 | 100.60 | 112.38 | 95.63 | 102.37 | **104.66** |

Moreover, the **Antiberty-BO** model replaces the GLM module with a different antibody-specific transformer language model to gauge its impact on active learning efficiency.

### 5.3   Result analysis

#### 5.3.1   Analysis of GLMAb-BO performance

The comparison results of different methods are presented in Figure 2 and Table 1, highlighting notable findings. Firstly, batch-mode optimization methods (such as PEX and BatchBO) outperform non-batch-mode methods (like AntBO) in terms of discovering sequences with higher fitness scores. This advantage stems from the inherent diversity introduced by considering multiple sequences simultaneously in batch mode optimization. In contrast, non-batch mode methods are more susceptible to being trapped in local optima due to their limited diversity. Additionally, the utilization of GLMAb to filter the extensive sequence optimization space facilitates the exploration process, enabling the identification of optimal sequences within a few rounds. Moreover, leveraging feature embedding pretrained from the GLMAb model enhances the performance of the surrogate model in predicting fitness scores for unknown sequences, even with limited training data.

#### 5.3.2   Analysis of submodule performance

For the second question, the comparison results with different ablative methods are shown in Figure 2 and detailed in Table 2. We find GLMAb-BO to perform better than Antiberty-BO in the first few rounds, which indicates our pretrained GLMAb model's ability to filter out more sequences with unsatisfying naturalness. Meanwhile, we can find that with the help of the embedding feature from GLMAb, the performance of GLMAb-BO is better than GLMAb(w/o emb)-BO on most datasets.

By comparing GLMAb-BO with GLMAb-select and GLMAb-random, we can find that they have similar performance in the first few rounds thanks to the pre-trained GLM. However, given more rounds, GLMAb-BO can find the sequence with the overall best fitness score which indicates that our whole exploration framework can be helpful for exploring sequences with better naturalness. By comparing only GLMAb-select and GLMAb-random, we can find that with the help of the trained surrogate model, it can also greedily improve the searched sequence naturalness since it could have overall better fitness in the last few rounds.

## 6   Conclusion

In conclusion, we have presented an efficient and risk-aware antibody design framework that combines the power of protein language models and batch Bayesian optimization. Our approach addresses the challenges of time-consuming and expensive experimentation by leveraging predictive models to generate candidate sequences with higher naturalness and employing Bayesian optimization to explore the sequence space effectively. By incorporating uncertainty estimates into the acquisition function, our framework achieves a balance between exploration and exploitation, resulting in the identification of promising antibody candidates. Through extensive experiments on benchmark datasets, we have demonstrated the effectiveness of our method. Our framework surpasses state-of-the-art approaches in terms of both efficiency and the quality of designed sequences. By reducing the cost and time required for antibody design, our framework has the potential to expedite the discovery of new antibodies and contribute to advancements in the field.

## Acknowledgement

We would like to acknowledge Xingyi Cheng from Biomap provides the GLM model.

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

# Appendix

## Training of protein language model

Since pretraining General language models (GLM) [9] pretrained on natural languages have achieved noteworthy performance, we leveraged the GLM framework to train a language model of antibodies with 1 billion parameters (GLM-Ab). Specifically, GLM-Ab is trained on both the understanding (in-place token prediction) and the generation (next token prediction) tasks, which contain a blank filling task, a recovering random masked span task and a recovering CDR deleted region task. The model is trained on Observed Antibody Space [17] with a max length of 1024, 230K steps, and 2048 samples per batch. Other hyperparameters are the same with the official implementation of GLM [9].

Following [20, 4, 11], we utilize the perplexity (PPL) given by a protein language model to predict the fitness of proteins. The main training scheme and hyper-parameters are following [7].

## Correlation evaluation of protein language model with the CDR3 antibody candidate pool

To shed light on the relevance of the pre-filtering with our pre-trained protein language model. We plot the correlation between the predicted value and ground truth value on the candidate pool datasets using different protein language models. As demonstrated from Figure 4 and 5, we can find that our pre-trained GLM-Ab can have a better correlation than Antiberty, which makes it becomes more useful for pre-filtering out the sequence with low naturalness. However, we still find that the correlation is not quite high even lower than 0.5 which can validate that the BO model for sequence exploration is very necessary.

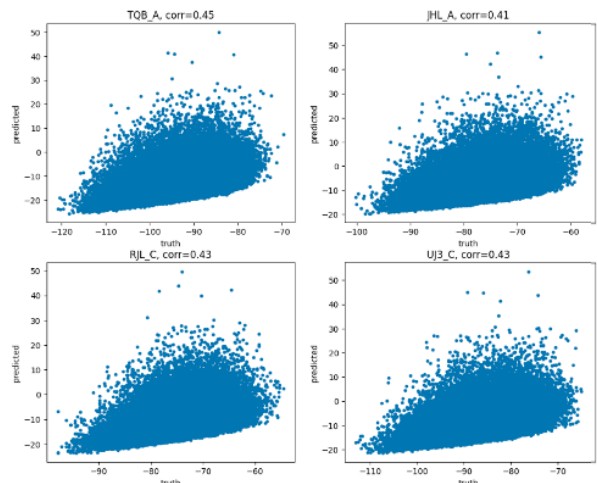

Figure 4: Correlation analysis between GLM-Ab and the candidate pool.

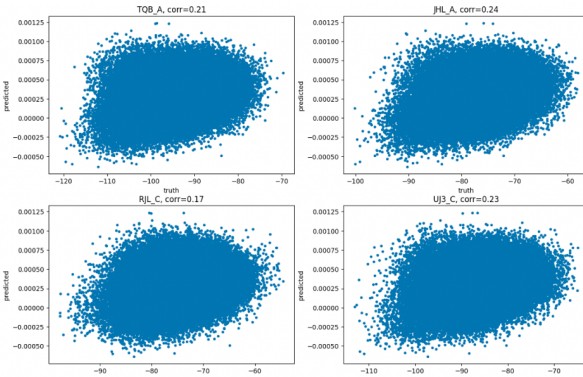

Figure 5: Correlation analysis between Antiberty and the candidate pool.

## Training of surrogate model

Constructing a surrogate model to facilitate the selection of mutants in in-silico evolutionary processes is an effective approach to mitigate the resource-intensive nature of wet-lab experiments. This involves training a fitness model denoted as $\hat{f}\theta$, where $\theta$ represents the model's parameters, to predict the fitness of mutant sequences. Specifically, the surrogate model is optimized by minimizing the regression loss function $L(\theta) = \mathbb{E}s \sim D\left[\left(\hat{f}\theta(s) - f(s)\right)^2\right]$, where $D$ signifies a dataset containing experimentally measured sequences. The acquired surrogate model $\hat{f}\theta$ becomes capable of predicting the fitness of previously unseen sequences, thereby guiding in-silico sequence exploration and enhancing the efficiency of directed evolution while reducing the need for extensive experimental efforts. Built upon the above trained GLM-Ab model's embedding, we add 6 layers of CNN module which is adapted from [26].

## Baseline methods setup

- **Combinatorial Bayesian Optimisation for Antibody Design (antbo):** Khan et al introduced a combinatorial Bayesian optimization framework for efficient *in silico* design of the CDRH3 region of antibodies. They used a CDRH3 trust region to restrict the search to sequences with favorable developability scores and a black-box oracle to score target specificity and affinity. However, it could only propose one sequence in each round of optimization. We adapt this method to propose 100 sequences to make a fair comparison.

- **Proximal Exploration(pex):** Ren et al proposed the Proximal Exploration (PEX) algorithm and the Mutation Factorization Network (MuFacNet) architecture for machine learning-guided protein sequence design. The PEX algorithm prioritizes the search for high-fitness mutants with low mutation counts, leveraging the natural property of the protein fitness landscape that a concise set of mutations upon the wild-type sequence are usually sufficient to enhance the desired function. The MuFacNet architecture is designed to predict low-order mutational effects, improving the sample efficiency of model-guided evolution.

- **Batch Bayes Optimization (batchbo):** We follow the idea from [5], and we apply the neural network ensemble with uncertainty estimate on the batch of sequence and use expected improvement as the acquisition function.

- **Random Search:** This method involves randomly selecting a subset of sequences from a larger pool, with the goal of establishing a reference point against which the performance of other methods can be compared. While this approach is simple, it can be useful for identifying cases where more sophisticated algorithms may be necessary. However, the quality of the baseline can be highly dependent on the selection method and the size of the subset. Therefore, care must be taken in the selection process to ensure that the resulting subset is representative of the larger pool of sequences. Overall, random selection can provide a valuable starting point for evaluating the performance of more advanced algorithms in a variety of bioinformatics applications.

## Ablative study methods setup

For the ablative study, we aim to evaluate the effectiveness of our proposed improvements. We construct several ablative versions based on our proposed GLMAb-BO method. We construct the following baselines:

- **GLMAb-score:** for this method, we only report the highest predicted score generated by GLMAb on our raw candidate pool $\mathcal{D}$.

- **GLMAb-select:** for this model, we eliminate the acquisition function, i.e., the evaluation function from Equation 6. And we only use the surrogate model to select the top sequence for the query.

- **GLMAb-random:** for this model, we eliminate both the acquisition function, i.e., the evaluation function from Equation 6 and the surrogate model. We only use the GLM model to filter out the sequence with worse scores. Then, we use a random method to select the top 100 query sequences.

- **GLMAb(w/o emb)-BO:** for this model, we only eliminate the GLMAB's embedding on top of the CNN surrogate model to test the effectiveness of the feature embedding module.

- **Antiberty-BO:** To evaluate the effectiveness of our proposed method's GLM module for active learning, we also tried another antibody-specific transformer language model [24] to replace the GLM module used before.

The detailed ablative methods' configuration summarization is summarized in Table 3.

Table 3: Comparison of the configuration of different ablation study methods

| Method | PLM | Surrogate model | Acquisition function |
|---|---|---|---|
| GLMAb-score | GLMAb | × | × |
| GLMAb-random | GLMAb | × | random |
| GLMAb-select | GLMAb | GLMAb emb+CNN | × |
| GLMAb(w/o emb)-BO | GLMAb | CNN | BO |
| Antiberty-BO | Antiberty | Antiberty emb+CNN | BO |
| GLMAb-BO (full model) | GLMAb | GLMAb emb+CNN | BO |

