# OpenReview forum: "Sample-efficient Antibody Design through Protein Language Model for Risk-aware Batch Bayesian Optimization"
_NeurIPS.cc/2023/Workshop/AI4Science — NeurIPS2023-AI4Science Poster_

### Official Review · Reviewer_njzt · 2023-10-23
**Review of Sample-efficient Antibody Design through Protein Language Model for Risk-aware Batch Bayesian Optimization**

**Rating:** 7
**Confidence:** 4

**Review:**

This paper proposes GLMAb-BO, an antibody design framework that leverages a language model to generate candidates and a Bayesian optimization to select from them. Using Absolut! as a computational ground truth, the proposed framework shows an advantageous performance compared to the selected baselines. I think it is reasonable to accept this paper to the workshop.

---

### Official Review · Reviewer_ugP2 · 2023-10-24
**Ok paper which seems to ask (and answer) the wrong questions: leaning towards rejection**

**Rating:** 4
**Confidence:** 3

**Review:**

This paper proposes a Bayesian optimization method to optimize protein sequences. However, I found the motivation for the method and its empirical evaluation to be a bit strange.

The authors start their motivation by saying:

> However, using traditional Gaussian processes (22) to measure the uncertainty for a large sequence is extremely inefficient.

This does not seem correct to me: in general, the computational complexity of GPs scales linearly with the input size and cubically with the number of data points (unless a more complicated kernel is used). At the very least, if a common kernel like RBF or Matern is used then the scaling is just linear with the sequence size.

Regardless, how their ensemble separates aleatoric or epistemic uncertainty is not clearly explained (I have not read the "PEX" paper that they cited). The main algorithmic contribution, the "risk aware" acquisition function, seems to just be a re-scaled version of UCB: that is, it is equivalent to just setting a specific value for $\beta$ (chosen by adjusting an initial ad-hoc choice by a formula which is also ad-hoc). It is unclear if/why this perspective is different/better than just tuning $\beta$ directly. I surmise that the proposed method is not too much different than just using variation in the dataset to select a good value for UCB's $\beta$.

The experiments in section 5 are a good start, but don't seem to answer the questions which I think are most important. First, the authors compare a bunch of methods from previous papers to their own method. The performance of all methods seems fairly similar and no metrics of variation are reported, but the authors conclude that their own method works best. To me this experiment does not have much value because 1) the random variation is not reported 2) it is unclear how, if at all, the hyperparameters of all the methods were tuned, and 3) the baselines use a mix of different surrogate models and acquisition functions, making it unclear what exactly causes the performance difference by the authors' proposed method. The ablation study in section 5.2 suggests that the acquisition function does not actually make a huge difference, and the high performance of "GLMAb-select" suggests that maybe it is the language model which is responsible for most of the performance gains.

Overall, I think this paper has potential, but in its current form the scientific insights are fairly marginal. Therefore I will give a score of 4, but I am hopeful that the authors can improve the manuscript in the future.

---

### Meta-Review · Area_Chair_3RbW · 2023-10-27

**Recommendation:** Accept (Poster)
**Confidence:** 4

**Metareview:**

&nbsp;

I'm delighted to recommend acceptance of the paper to the workshop. The reviewers mentioned a number of points that are worth addressing ahead of the workshop:

&nbsp;

1. It would be great to have some discussion of the connections between the proposed risk-aware acquisition function and UCB as suggested by Reviewer ugP2. It would also be interesting to understand how the risk-aware acquisition relates to approaches that aim to penalize aleatoric noise in the suggestions (risk-averse BO) [1, 2].
2. In terms of the applicability of Gaussian processes to sequence data, the subsequence string kernel (SSK) [3, 4] does indeed suffer from a complexity standpoint when faced with long sequences. It would be worth expanding on this in the paper. Additionally, it may be worth adding a "Bag of Amino Acids" baseline as a sanity check cf. the following [tutorial](https://github.com/leojklarner/gauche/blob/main/notebooks/Protein%20Fitness%20Prediction%20-%20Bag%20of%20Amino%20Acids.ipynb) from [5]. In this instance, the computational complexity of using the GP-SSK approach with long sequences would be avoided.
3. It would indeed be beneficial to report some measure of statistical variation in Tables 1 and 2.

&nbsp;

I look forward to discussing this work in more detail at the workshop!

&nbsp;

__**References**__

&nbsp;

[1] Griffiths, R.R., Aldrick, A.A., Garcia-Ortegon, M. and Lalchand, V., 2021. Achieving robustness to aleatoric uncertainty with heteroscedastic Bayesian optimisation. Machine Learning: Science and Technology, 3(1), p.015004.

[2] Makarova, A., Usmanova, I., Bogunovic, I. and Krause, A., 2021. Risk-averse heteroscedastic Bayesian optimization, NeurIPS 2021.

[3] Moss, H., Leslie, D., Beck, D., Gonzalez, J. and Rayson, P., 2020. BOSS: Bayesian optimization over string spaces. NeurIPS 2020.

[4] Beck, D. and Cohn, T., 2017, Learning kernels over strings using Gaussian processes. EMNLP 2017.

[5] Griffiths, R.R et al., 2022. GAUCHE: A library for Gaussian processes in chemistry. NeurIPS 2023.

&nbsp;